# Humanly Certifying Superhuman Classifiers

**Qiongkai Xu** [*]
University of Melbourne
Victoria, Australia
`qiongkai.xu@unimelb.edu.au`

**Christian Walder** [*]
Google Brain
Montreal, Canada
`cwalder@google.com`

**Chenchen Xu** [*]
Amazon
Canberra, Australia
`xuchench@amazon.com`

## Abstract

This paper addresses a key question in current machine learning research: if we believe that a model's predictions might be better than those given by human experts, how can we (humans) verify these beliefs? In some cases, this "superhuman" performance is readily demonstrated; for example by defeating top-tier human players in traditional two player games. On the other hand, it can be challenging to evaluate classification models that potentially surpass human performance. Indeed, human annotations are often treated as a ground truth, which implicitly assumes the superiority of the human over any models trained on human annotations. In reality, human annotators are subjective and can make mistakes. Evaluating the performance with respect to a genuine oracle is more objective and reliable, even when querying the oracle is more expensive or sometimes impossible. In this paper, we first raise the challenge of evaluating the performance of both humans and models with respect to an oracle which is *unobserved*. We develop a theory for estimating the accuracy compared to the oracle, using only imperfect human annotations for reference. Our analysis provides an executable recipe for detecting and certifying superhuman performance in this setting, which we believe will assist in understanding the stage of current research on classification. We validate the convergence of the bounds and the assumptions of our theory on carefully designed toy experiments with known oracles. Moreover, we demonstrate the utility of our theory by meta-analyzing large-scale natural language processing tasks, for which an oracle does not exist, and show that under our mild assumptions a number of models from recent years have already achieved superhuman performance with high probability—suggesting that our new oracle based performance evaluation metrics are overdue as an alternative to the widely used accuracy metrics that are naively based on imperfect human annotations.

## 1 Introduction

Artificial Intelligence (AI) agents have begun to outperform humans on remarkably challenging tasks; AlphaGo defeated top ranked Go players (Silver et al., 2016; Singh et al., 2017), and OpenAI's Dota2 AI has defeated human world champions of the game (Berner et al., 2019). These AI tasks may be evaluated objectively, *e.g.*, using the total score achieved in a game and the victory against another player. However, for supervised learning tasks such as image classification and sentiment analysis, certifying a machine learning model as superhuman is subjectively tied to human judgments rather than comparing with an oracle. We focus on paving a way towards evaluating models with potentially superhuman performance in classification.

When evaluating the performance of a classification model, we generally rely on the accuracy of the predicted labels with regard to ground truth labels, which we call the *oracle accuracy*. However, or-

---

[*]Work was done while the authors were with the Australian National University and Data61 CSIRO.

acle labels may arguably be unobservable. For tasks such as object detection and saliency detection, the predictions are subjective to many factors of the annotators, *e.g.*, their background and physical or mental state. For other tasks, even experts may not be able to summarize an explicit rule for the prediction, such as predicting molecule toxicity and stability. Without observing oracle labels researchers often resort to two heuristics, *i)* human predictions or aggregated human annotations are effectively treated as ground truth (Wang et al., 2018; Lin et al., 2014; Wang et al., 2019) to approximate the oracle, and *ii)* the *inter-annotator aggreement* is taken as the best possible machine learning model performance (for an extensive survey of works that make this claim without proof, see the works cited within (Boguslav & Cohen, 2017; Richie et al., 2022)). This heuristic approach suffers some key disadvantages. Firstly, the quality control of human annotation is challenging (Artstein, 2017; Lampert et al., 2016). Secondly, current evaluation paradigms focus on evaluating the performance of models, but not the oracle accuracy of humans — yet we cannot claim that a machine learning model is superhuman without properly estimating the human performance as compared to the oracle. Thirdly, as machine learning models exceed human performance on important tasks, it becomes insufficient to merely report the agreement of the model to human annotations.

In this paper, we work on the setting that oracle labels are unobserved (see Figure 1). Within this setting is provided a theory for estimating the oracle accuracy on classification tasks which formalises what empirical works have hinted towards (Richie et al., 2022), that machine learning classification models may outperform the humans who provide them with training supervision. Our aim is not to optimally combine machine learning systems, but rather to estimate the *oracle* accuracy of a single machine learning system by comparing it with the results obtained from multiple human annotators. Our theory includes *i)* upper bounds for the averaged oracle accuracy of the annotators, *ii)* lower bounds for the oracle accuracy of the model, and *iii)* finite sample analysis for both bounds and their margin which represents the model's outperformance.

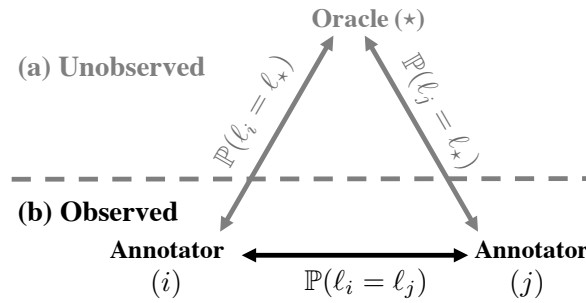

Figure 1: The relationship between *a)* the oracle accuracy of the annotators, $\mathbb{P}(\ell_i = \ell_\star)$, and *b)* the agreement between two annotators, $\mathbb{P}(\ell_i = \ell_j)$. $\ell_i$ and $\ell_j$ are labels given by annotator $i$ and $j$, $\ell_\star$ is the oracle label. In our setting, part *a)* is unobserved (*gray*) and part *b)* is observed (*black*).

represents the model's outperformance. Based on our theory, we propose an algorithm to detect competitive models and to report confidence scores, which formally bound the probability that a given model outperforms the average human annotator. Empirically, we observe that some existing models for sentiment classification and natural language inference (NLI) have already achieved superhuman performance with high probability.

## 2 EVALUATION THEORY

We now present our theory for human annotators and machine learning models with oracle labels.

### 2.1 PROBLEM STATEMENT

We are given $K$ labels crowd sourced from $K$ human annotators, $\{\ell_i\}_{i=1}^K$, and some labels from a model $\ell_\mathcal{M}$. The probability of two annotators $a_i$ and $a_j$ possess matched annotations with the other is $\mathbb{P}(\ell_i = \ell_j)$. Denote by $\ell_\mathcal{K}$ the label of the "average" human annotator which we define as the label obtained by selecting one of the $K$ human annotators uniformly at random. We seek to formally compare the oracle accuracy of the average human, $\mathbb{P}(\ell_\mathcal{K} = \ell_\star)$, with that of the machine learning model, $\mathbb{P}(\ell_\mathcal{M} = \ell_\star)$, where $\ell_\star$ is the unobserved oracle label. Denote by $\ell_\mathcal{G}$ the label obtained by aggregating (say, by voting) the $K$ human annotators' labels. We distinguish between the oracle accuracy $\mathbb{P}(\ell_\mathcal{M} = \ell_\star)$ and the agreement with human annotations $\mathbb{P}(\ell_\mathcal{M} = \ell_\mathcal{G})$, although these two concepts have been confounded in many previous applications and benchmarks.

## 2.2 An Upper Bound for the Average Annotator Performance

The oracle accuracy of the average annotator $\ell_{\mathcal{K}}$ follows the definition of the previous section, and conveniently equals the average of the oracle accuracy of each annotator, *i.e.*

$$\mathbb{P}(\ell_{\mathcal{K}} = \ell_{\star}) = \frac{1}{K} \sum_{i=1}^{K} \mathbb{P}(\ell_i = \ell_{\star}). \tag{1}$$

By introducing an assumption as equation 2, we may bound the above quantity. Intuitively, annotators are likely to be positively correlated because *i)* they tend to have the same correct or wrong annotations on the same easy or difficult tasks respectively, *ii)* they may share similar backgrounds that affect their decisions, and *etc*. Note that this assumption is also discussed in Section 3.2 (see RQ2) where we provide supporting evidence for it on a real-world problem with known oracle labels.

**Theorem 1 (Average Performance Upper Bound)** *Assume annotators are positively correlated,*

$$\mathbb{P}(\ell_i = \ell_{\star} | \ell_j = \ell_{\star}) \geq \mathbb{P}(\ell_i = \ell_{\star}). \tag{2}$$

*Then, the upper bound of averaged annotator accuracy with respect to the oracle is*

$$\mathbb{P}(\ell_{\mathcal{K}} = \ell_{\star}) \leq \mathcal{U} \triangleq \sqrt{\frac{1}{K^2} \sum_{i=1}^{K} \sum_{j=1}^{K} \mathbb{P}(\ell_i = \ell_j)}. \tag{3}$$

We observe that average inter-annotator agreement will be over-estimated by including the self comparison terms $\mathbb{P}(l_i = l_j)$, which is always equal to one when $i = j$, but that the total overestimation to $\mathcal{U}^2$ is less or equal to $1/K$ ($K$ out of $K^2$ terms), and that the influence will reduce and converge to zero as limit $K \to \infty$. To provide a more practical estimation, we introduce an empirically approximated upper bound $\mathcal{U}^{(e)}$. In contrast, $\mathcal{U}$ in equation 3 is also noted as theoretical upper bound, $\mathcal{U}^{(t)}$.

**Definition 1** *The **empirically approximated upper bound**,*

$$\mathcal{U}^{(e)} \triangleq \sqrt{\frac{1}{K(K-1)} \sum_{i=1}^{K} \sum_{\substack{j=1 \\ i \neq j}}^{K} \mathbb{P}(\ell_i = \ell_j)}. \tag{4}$$

**Lemma 2 (Convergence of $\mathcal{U}^{(e)}$)** *Assume that $\sum_{j=1, i \neq j}^{K} \mathbb{P}(\ell_i = \ell_j) \geq \frac{K-1}{N_c}$, where $N_c$ is the constant number of classes. The approximated upper bound $\mathcal{U}^{(e)}$ satisfies*

$$\lim_{K \to +\infty} \mathcal{U}/\mathcal{U}^{(e)} = 1. \tag{5}$$

*Therefore, with large $K$, $\mathcal{U}^{(e)}$ converges to $\mathcal{U}$ or $\mathcal{U}^{(t)}$.*

Empirical support for the convergence of $\mathcal{U}^{(e)}$ to $\mathcal{U}^{(t)}$ are demonstrated in Figure 3 of Section 3.2.

## 2.3 A Lower Bound for Model Performance

For our next result, we introduce another assumption as equation 6. Given two predicted labels $\ell_a$ and $\ell_b$, we assume that $\ell_b$ is reasonably predictive even on those instances that $a$ gets wrong, as per the assumption formally stated within the following theorem. Note that this assumption is rather mild in that even random guessing satisfies it, as in this case the probability of choosing the correct label is equal to any other single wrong label. Once again, this key assumption is discussed and validated on human data with known oracle labels in Section 3.2 (see RQ2).

**Theorem 3 (Performance Lower Bound)** *Assume that for any single incorrect label $\ell_{\times} \neq \ell_{\star}$,*

$$\mathbb{P}(\ell_b = \ell_{\star} | \ell_a \neq \ell_{\star}) \geq \mathbb{P}(\ell_b = \ell_{\times} | \ell_a \neq \ell_{\star}). \tag{6}$$

*Then, the lower bound for the oracle accuracy of $\ell_b$ is*

$$\mathcal{L} \triangleq \mathbb{P}(\ell_a = \ell_b) \leq \mathbb{P}(\ell_b = \ell_{\star}). \tag{7}$$

In practice, a more accurate $\ell_a$ gives a tighter lower bound for $\ell_b$, and so we employ the aggregated human annotations for the former (letting $\ell_a = \ell_{\mathcal{G}}$) to calculate the lower bound of the machine learning model (letting $\ell_b = \ell_{\mathcal{M}}$), as demonstrated in Section 3.2.

**Connection to traditional practice of accuracy calculation.** Generally, the ground truth of a benchmark corpus is constructed by aggregating multiple human annotations (Wang et al., 2018; 2019). For example, the averaged sentiment score is used in SST (Socher et al., 2013) and majority of votes in SNLI (Bowman et al., 2015). Then, the aggregated annotations are treated as ground truth to calculate accuracy. Under this setting, the 'traditional' accuracy score evaluated on the (aggregated) human ground truth can be viewed as a special case of our lower bound.

## 2.4    FINITE SAMPLE ANALYSIS

The results above assume that the agreement probabilities are known; we now connect them with the finite sample case: $\ell^{(n)}$ denotes the label assigned to the $n$-th data point in accordance to $\ell$, for $n = 1, 2, \ldots, N$. $\mathbb{P}^{(N)}$ is the empirical probability given $N$ observations, and $\mathbb{P}$ is $\lim_{N \to \infty} \mathbb{P}^{(N)}$. We begin with a standard concentration inequality (see *e.g.* (Boucheron et al., 2013, § 2.6)),

**Theorem 4 (Hoeffding's Inequality)** *Let $X_1, \ldots, X_N$ be independent random variables with finite variance such that $\mathbb{P}(X_n \in [\alpha, \beta]) = 1$, for all $1 \le n \le N$. Let*

$$\overline{X} \triangleq \frac{1}{N} \sum_{n=1}^{N} X_n,$$

*then, for any $t > 0$,*

$$\mathbb{P}(\overline{X} - \mathbb{E}[\overline{X}] \ge +t) \le \exp\left(-\frac{2Nt^2}{(\alpha - \beta)^2}\right),$$

$$\mathbb{P}(\overline{X} - \mathbb{E}[\overline{X}] \le -t) \le \exp\left(-\frac{2Nt^2}{(\alpha - \beta)^2}\right). \tag{8}$$

Combining this with Thereom 1 we obtain the following.

**Theorem 5 (Sample Average Performance Upper Bound)** *With Theorem 1's assumptions and*

$$\mathbb{P}^{(N)}(\ell_i = \ell_j) = \frac{1}{N} \sum_{n=1}^{N} \left[ \ell_i^{(n)} = \ell_j^{(n)} \right] \tag{9}$$

*defining the empirical agreement ratio,[1] and letting*

$$\delta_u = \exp(-2Nt_u^2). \tag{10}$$

*With probability at least $1 - \delta_u$, for any $t_u > 0$,*

$$\mathbb{P}(\ell_{\mathcal{K}} = \ell_\star) \le \sqrt{t_u + \frac{1}{K^2} \sum_{i=1}^{K} \sum_{j=1}^{K} \mathbb{P}^{(N)}(\ell_i = \ell_j)}. \tag{11}$$

**Theorem 6 (Sample Performance Lower Bound)** *With Theorem 3's assumptions and equation 9, define*

$$\delta_l = \exp(-2Nt_l^2). \tag{12}$$

*With probability at least $1 - \delta_l$, for any $t_l > 0$,*

$$\mathbb{P}^{(N)}(\ell_a = \ell_b) \le t_l + \mathbb{P}(\ell_b = \ell_\star). \tag{13}$$

## 2.5    DETECTING AND CERTIFYING SUPERHUMAN MODELS

We propose a procedure to discover potentially superhuman models based on our theorems.

1. Calculate the upper bound of the average oracle accuracy of human annotators, $\mathcal{U}_N$, with $N$ data samples;

---

[1] Here $[\cdot]$ is the Iverson bracket.

2. Calculate the lower bound of the model oracle accuracy $\mathcal{L}_N$ using aggregated human annotations as the reference[2], with $N$ data samples;

3. Check whether the finite sample margin between the bounds $\mathcal{L}_N - \mathcal{U}_N$ is larger than zero;[3]

4. Give proper estimation of $t_u$ and $t_l$ and calculate a confidence score of $\mathbb{P}(\mathcal{L} - \mathcal{U} \geq 0)$.

Generally, larger margin indicates higher confidence of out-performance. To formally check confidence for the aforementioned margin we provide the following theorem and corresponding algorithms.

**Theorem 7 (Confidence of Out-Performance)** *Assume an annotator pool with agreement statistic* $\mathcal{U}_N$ *of equation 34, and an agreement statistic between model and aggregated annotations* $\mathcal{L}_N$ *of equation 39. If* $\mathcal{L}_N > \mathcal{U}_N$ *then for all* $\tau \geq 0$, $t_u \geq 0$ *and* $t_l \geq 0$ *that satisfy*

$$\mathcal{L}_N - t_l - \sqrt{t_u + \mathcal{U}_N^2} = \tau, \tag{14}$$

*with probability at least* $1 - \delta_u - \delta_l$*, the oracle accuracy of the model exceeds that of the average annotator by* $\tau$*,*

$$\mathbb{P}\big(\mathbb{P}(\ell_{\mathcal{M}} = \ell_\star) - \mathbb{P}(\ell_{\mathcal{K}} = \ell_\star) \geq \tau\big) \geq 1 - \delta_l - \delta_u, \tag{15}$$

*where*

$$\delta_u = \exp\left(-2Nt_u^2\right), \delta_l = \exp\left(-2Nt_l^2\right). \tag{16}$$

**Confidence Score Estimation.** The above theorem suggests the confidence score

$$S = 1 - \delta_l - \delta_u, \tag{17}$$

and we need only choose the free constants $t_l, t_u$ and $\tau$. Recall equation 14,

$$\tau = (\mathcal{L}_N - t_l) - \sqrt{t_u + \mathcal{U}_N^2}, \tag{18}$$

and remove one degree of freedom parameterise in $t_u$ as

$$t_l(t_u, \tau) = \mathcal{L}_N - \tau - \sqrt{t_u + \mathcal{U}_N^2}. \tag{19}$$

We are interested in $\mathbb{P}(\mathcal{L} - \mathcal{U} \geq 0)$ so we choose $\tau = 0$, and give two choices for $t_u$ and $t_l$.

**Algorithm 1 (Heuristic Margin Separation, HMS).** We assign half of the margin to $t_u$,

$$t_u = \frac{\mathcal{L}_N - \mathcal{U}_N}{2}. \tag{20}$$

Then, with $\tau = 0$ we calculate the corresponding

$$t_l = \mathcal{L}_N - \sqrt{\frac{\mathcal{L}_N - \mathcal{U}_N}{2} + \mathcal{U}_N^2}, \tag{21}$$

and compute the heuristic confidence score $S$.

**Algorithm 2 (Optimal Margin Separation, OMS).**
For a locally (in $t_u$) optimal score, we perform gradient ascent (Lemaréchal, 2012) on $S(t_u)$, where

$$S(t_u) = 1 - \delta(t_u) - \delta(t_l(t_u, 0)), \tag{22}$$

with $t_u$ is initialized as $(\mathcal{L}_N - \mathcal{U}_N)/2$ before optimization[4].

---

[2]We demonstrate that aggregating the predictions by voting and weighted averaging are effective in improving our bounds. We emphasize however that the aggregated predictions need not be perfect, as we do not assume that this aggregation yields an oracle.

[3]A larger deviation, say a high positive value, is of more interest to our certification as it gives a higher confidence score to the outperformance.

[4]We set the learning rate to 1e-4, and iterated 100 times.

## 3    EXPERIMENTS AND DISCUSSION

Previously, we introduced a new theory for analyzing the oracle accuracy of set of classifiers using observed agreements between them. In this section, we demonstrate our theory on several classification tasks, to demonstrate the utility of the theory and reliability of the associated assumptions. Our code is available at `https://github.com/xuqiongkai/Superhuman-Eval.git`.

### 3.1    EXPERIMENTAL SETUP

We first consider two classification tasks with oracle labels generated by rules. Given the oracle predictions, we are able to empirically validate the assumptions for our theorems and observe the convergence of the bounds. Then, we apply our theory on two real-world classification tasks and demonstrate that some existing state-of-the-art models have potentially achieved better performance than the (averaged) performance of the human annotators in reference to the (unobserved) oracle.

**Classification tasks with oracle rules.**    To validate the correctness of our theory, we collect datasets with observable oracle labels. We construct two visual cognitive tasks, **Color Classification** and **Shape Classification**, with explicit unambiguous rules to acquire oracle labels, as follows:

- **Color Classification**: select the most frequently occurring color of the objects in an image.
- **Shape Classification**: select the most frequently occurring shape of the objects in an image.

For both tasks, object size is ignored. As illustrated in Figure 2, we vary colors (*Red*, *Blue* and *Yellow*) and shapes (*Triangle*, *Square*, *Pentagon*, *Hexagon* and *Circle*) for the two tasks, respectively.

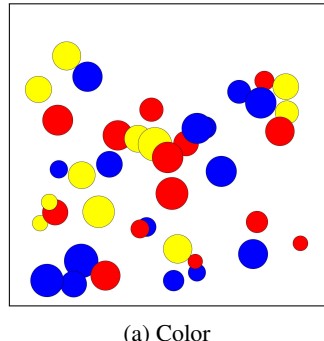
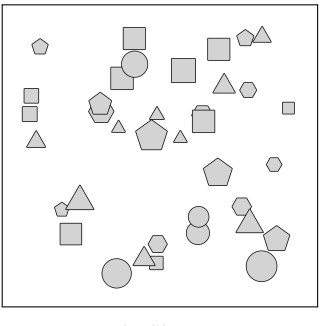

(a) Color

(b) Shape

Figure 2: Example *a)* **Color Classification** and *b)* **Shape Classification**. *a)* includes 40 objects of three colors, *Red* (14), *Blue* (15) and *Yellow* (11), with *Blue* as the most frequent color and therefore the oracle label. *b)* includes 37 objects of five different shapes, *Triangle* (9), *Square* (10), *Pentagon* (7), *Hexagon* (6) and *Circle* (5), with *Square* the dominant shape and oracle label.

For each task, we generated 100 images and recruited 10 annotators from the *Amazon Mechanical Turk*[5] to label them. Each randomly generated example includes 20 to 40 objects. We enforce that no objects overlap more than 70% with all others, and that there is only one class with the highest count, to ensure uniqueness of the oracle label. The oracle number of the colors and shapes are recorded to generate oracle labels of the examples. Note that our $\mathcal{U}^2$ is the average agreement among annotators, and so is proportional to Cohen's Kappa coefficient which we report in Appendix D along with additional details about the guidelines and interface presented to the annotators.

**Real-World Classification Tasks.**    We analyze the performance of human annotators and machine learning models on two real-world NLP tasks, namely sentiment classification and natural language inference (NLI). We use the Stanford Sentiment Treebank (**SST**) (Socher et al., 2013) for sentiment classification. The sentiment labels are mapped into two classes (SST-2)[6] or five classes (SST-5),

---

[5]`https://www.mturk.com`

[6]Samples with overall neutral scores are excluded as in (Tai et al., 2015).

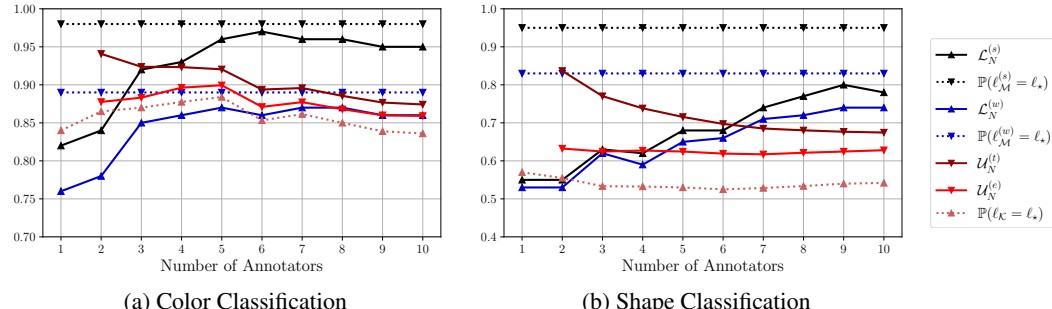

(a) Color Classification        (b) Shape Classification

Figure 3: Comparison of sample lower bound $\mathcal{L}_N$ for model oracle accuracy $\mathbb{P}(\ell_\mathcal{M} = \ell_\star)$. Relatively strong and weak models are indicated by $\mathcal{M}^{(s)}$ and $\mathcal{M}^{(w)}$. The aggregation of one annotator is based on the labels provided by the single annotator. Another comparison of sample theoretical upper bound $\mathcal{U}_N^{(t)}$ and sample empirical upper bound $\mathcal{U}_N^{(e)}$ of average oracle accuracy of annotators $\mathbb{P}(\ell_\mathcal{K} = \ell_\star)$.

*very negative* $([0, 0.2])$, *negative* $((0.2, 0.4])$, *neutral* $((0.4, 0.6])$, *positive* $((0.6, 0.8])$, and *very positive* $((0.8, 1.0])$. We use the Stanford Natural Language Inference (**SNLI**) corpus (Bowman et al., 2015) for NLI. All samples are classified by five annotators into three categories, *i.e.* *Contradiction* (C), *Entailment* (E), and *Neutral* (N). More details of the datasets are reported in Appendix C. In the latter part of this section, we focus on the estimated upper bounds on test sets, as we intend to compare them with the performance of machine learning models generally evaluated on test sets.

**Machine Learning Models.** For both of the classification tasks with known oracles, we treat them as detection tasks and train YOLOv3 models (Redmon & Farhadi, 2018) for them. The input image resolution is $608 \times 608$, and we use the proposed Darknet-53 as the backbone feature extractor. For comparison, we train two models, a strong model and a weak model, on 512 and 128 randomly generated examples, respectively. All models are trained for a maximum of 200 epochs until convergence. During inference, the model detects the objects and we count each type of object to obtain the prediction. We compare several representative models and their variants for real-world classification tasks, such as Recurrent Neural Networks (Chen et al., 2018; Zhou et al., 2015), Tree-based Neural Networks (Mou et al., 2016; Tai et al., 2015), and Pre-trained Transformers (Devlin et al., 2019; Radford et al., 2018; Wang et al., 2020; Sun et al., 2020).

## 3.2 RESULTS AND DISCUSSION

We now conduct several experiments to validate the convergence of the bounds and the validity of the assumptions. We then demonstrate the utility of our theory by detecting real-world superhuman models. We organize the discussion into several research questions (RQ).

**RQ1: Will the bounds converge given more annotators?** We first analyze the lower bounds. We demonstrate lower bounds for strong (s) and weak (w) models in Figure 3 in black and blue lines respectively. Generally, *i)* lower bounds $\mathcal{L}_N$ are always under the corresponding oracle accuracy; *ii)* the lower bounds grow and tend to get closer to the bounded scores given more aggregated annotators. Then, we analyze the upper bounds. We illustrate theoretical upper bound $\mathcal{U}_N^{(t)}$ and empirically approximated upper bound $\mathcal{U}_N^{(e)}$, in comparison with average oracle accuracy of annotators $\mathbb{P}(\ell_\mathcal{K} = \ell_\star)$, in Figure 3. We observe that *i)* both upper bounds give higher estimation than the average oracle accuracy of annotators; *ii)* the margin between $\mathcal{U}_N^{(t)}$ and $\mathcal{U}_N^{(e)}$ reduce, given more annotators incorporated; *iii)* $\mathcal{U}_N^{(e)}$ generally provides a tighter bound than $\mathcal{U}_N^{(t)}$, and we will use $\mathcal{U}_N^{(e)}$ as $\mathcal{U}_N$ to calculate confidence score in later discussion.

**RQ2: Are the assumptions of our theorems valid?** We verify the key assumptions for the upper bound of Theorem 1 and the lower bound of Theorem 3 by computing the relevant quantities in Table 1. The assumptions within these theorems are not concerned with agreement (or otherwise) on particular training examples (which could be unrealistic), but rather are statements

Table 2: The sample theoretical upper bounds and sample empirically approximated upper bounds, $\mathcal{U}_N^{(t)}$ and $\mathcal{U}_N^{(e)}$, of the average oracle accuracy of the human annotators, and the sample lower bounds $\mathcal{L}_N$ of some representative models on the SST and SNLI tasks. Those models with $\mathcal{L}_N$ higher than $\mathcal{U}_N^{(e)}$ or even $\mathcal{U}_N^{(t)}$ are highlighted with † or ‡.

| B | SST 5-Class | | SST 2-Class | | SNLI 3-Class | |
|---|---|---|---|---|---|---|
| | Classifier | Score | Classifier | Score | Classifier | Score |
| $\mathcal{U}_N^{(t)}$ | Avg. Human | 0.790 ‡ | Avg. Human | 0.960 ‡ | Avg. Human | 0.904 ‡ |
| $\mathcal{U}_N^{(e)}$ | Avg. Human | 0.660 † | Avg. Human | 0.939 † | Avg. Human | 0.879 † |
| $\mathcal{L}_N$ | CNN-LSTM (Zhou et al., 2015) | 0.492 | CNN-LSTM (Zhou et al., 2015) | 0.878 | BiLSTM (Chen et al., 2018) | 0.855 |
| | Constituency Tree-LSTM (Tai et al., 2015) | 0.510 | Constituency Tree-LSTM (Tai et al., 2015) | 0.880 | Tree-CNN (Mou et al., 2016) | 0.821 |
| | BERT-large (Devlin et al., 2019) | 0.555 | BERT-large (Devlin et al., 2019) | 0.949† | LM-Pretrained Transformer (Radford et al., 2018) | 0.899† |
| | RoBERTa+Self-Explaining (Sun et al., 2020) | 0.591 | StructBERT (Wang et al., 2020) | 0.971‡ | SemBERT (Zhang et al., 2020) | 0.919‡ |

made in aggregate over all input data points. In words, Theorem 1 assumes that the probability that an annotator predicts the oracle label must increase when we assume that that any other annotator predicts the oracle label *on average, over all classifier inputs and class labels*.

Theorem 3 assumes that *on average, over all classifier inputs and class labels*, if the majority vote by the human is incorrect w.r.t. the oracle, then the machine learning model is still more likely to predict the oracle label *than any other specific label that disagrees with the oracle*. The two assumptions clearly hold in our specially designed experiments with real human subjects, although we can only perform this analysis on the tasks with known oracle labels. However, the methodology behind Table 1 is by design rather conservative, as we sum over all incorrect labels (see column 2 of Table 1.b). Despite this stricter setup, our assumption still holds on both experiments.

Table 1: Validating our assumptions for both upper bound Theorem 1 and lower bound Theorem 3 on Color and Shape.

| Task | $\mathbb{P}(\ell_i = \ell_\star \mid \ell_j = \ell_\star)$ | $\mathbb{P}(\ell_i = \ell_\star)$ |
|---|---|---|
| Color | **0.850** | 0.836 |
| Shape | **0.586** | 0.542 |

(a) Theorem 1 assumes $\mathbb{P}(\ell_i = \ell_\star \mid \ell_j = \ell_\star) \geq \mathbb{P}(\ell_i = \ell_\star)$, $i \neq j$

| Task b | $\mathbb{P}(\ell_b = \ell_\star \mid \ell_a \neq \ell_\star)$ | $\sum_{\ell_\times \neq \ell_\star} \mathbb{P}(\ell_b = \ell_\times \mid \ell_a \neq \ell_\star)$ |
|---|---|---|
| Color $\mathcal{M}^{(w)}$ | **1.000** | 0.000 |
| Color $\mathcal{M}^{(s)}$ | **1.000** | 0.000 |
| Shape $\mathcal{M}^{(w)}$ | **0.579** | 0.421 |
| Shape $\mathcal{M}^{(s)}$ | **0.895** | 0.105 |

(b) Theorem 3 assumes $\mathbb{P}(\ell_b = \ell_\star \mid \ell_a \neq \ell_\star) \geq \mathbb{P}(\ell_b = \ell_\times \mid \ell_a \neq \ell_\star)$

*Disclaimer:* while the assumptions appear reasonable, we recommend where possible to obtain some oracle labels to validate the assumptions when applying our theory.

**RQ3: How to identify a 'powerful', or even superhuman, classification model?** We first compare the $\mathcal{L}_N$ with $\mathcal{U}_N$ in our toy experiments, in Figure 3. Overall, showing superhuman performance is more likely given more annotators. $\mathcal{L}_N^{(s)}$ outperforms both $\mathcal{U}_N^{(e)}$ and $\mathcal{U}_N^{(t)}$, given more than 4 and 6 annotators for color classification and shape classification, respectively. When the model is marginally outperforming the humans, see weak model for color classification, we may not observe a clear superhuman performance margin, $\mathcal{L}_N^{(w)}$ and $\mathcal{U}_N^{(e)}$ are very close given more than 7 annotators.

For real-world classification tasks, we *i)* calculate the average annotator upper bounds given multiple annotators' labels and *ii)* collect model lower bounds reported in previous literature. Results on SST and SNLI are reported in Table 2; pretrained language models provide significant performance improvement on those tasks. Our theory identifies some of these

Table 3: Confidence score $S$ for the certificated models that outperform human annotators in SST-2 and SNLI.

| Model | Task | S(HMS) | S(OMS) |
|---|---|---|---|
| Devlin et al. (2019) | SST-2 | $< 0$ | $< 0$ |
| Wang et al. (2020) | SST-2 | 0.4730 | **0.6208** |
| Radford et al. (2018) | SNLI | 0.8482 | **0.9267** |
| Zhang et al. (2020) | SNLI | 0.9997 | **0.9999** |

models that exceed average human performance with high probability, by comparing $\mathcal{U}_N^{(e)}$ or the even more restrictive $\mathcal{U}_N^{(t)}$.

**RQ4: How confident are the certifications?** We calculate our confidence score for the identified outperforming models via $\mathcal{U}_N$, $\mathcal{L}_N$, $N$, and using HMS and OMS, as reported in Table 3. Generally, the confidence scores for SNLI models are higher than those of SST-2 because the former has test set is more than five times larger, while more recent and advanced models achieve higher confidence scores as they have larger margin of $\mathcal{L}_N - \mathcal{U}_N$.

## 4 RELATED WORK

Classification accuracy is a widely used measure of model performance (Han et al., 2011), although there are other options such as precision, recall, F1-score (Chowdhury, 2010; Sasaki et al., 2007), Matthews correlation coefficient (Matthews, 1975; Chicco & Jurman, 2020), *etc.*. Accuracy measures the disagreement between the model outputs and some reference labels. A common practice is to collect human labels to treat as the reference. However, we argue that the ideal reference is rather the (unobserved) oracle, as human predictions are imperfect. We focus on measuring the *oracle accuracy* for both human annotators and machine learning models, and for comparing the two.

A widely accepted approach is to crowd source (Kittur et al., 2008; Mason & Suri, 2012) a dataset for testing purposes. The researchers collect a large corpus with each examples labeled by multiple annotators. Then, the aggregated annotations are treated as ground truth labels (Socher et al., 2013; Bowman et al., 2015). This largely reduces the variance of the prediction (Nowak & Rüger, 2010; Kruger et al., 2014), however, such aggregated results are still not oracle, and their difference to oracle remains unclear. In our paper, we prove that the accuracy on aggregated human prediction, as ground truth, could be considered as a special case of the lower bound of oracle accuracy for machine learning models. On the other hand, much work considers the reliability of collected data, by providing the agreement scores between annotators (Landis & Koch, 1977). Statistical measures for the reliability of the inter-annotator agreement (Gwet, 2010), such as Cohen's Kappa (Pontius Jr & Millones, 2011) and Fleiss' Kappa (Fleiss, 1971), are normally based on the raw agreement ratio. However, the agreement between annotators does not obviously reflect the oracle accuracy; *e.g.* identical predictions from two annotators does not mean they are both oracles. In our paper, we prove that observed agreement between all annotators could serve as an upper bound for the average oracle accuracy of those annotators. Overall, we propose a theory for comparing the oracle accuracy of human annotators and machine learning models, by connecting the aforementioned bounds.

The discovery that models can predict better than humans dates back at least to the seminal work (Meehl, 1954), which compared *ad hoc* predictions based on subjective information, to those based on simple linear models with a (typically small) number of relevant numeric attributes. Subsequent work found that one may even train such a model to mimic the predictions made by the experts (rather than an oracle), and yet still maintain superior out of sample performance (Goldberg, 1970). The comparison of human and algorithmic decision making remains an active topic of psychology research (Kahneman et al., 2021). Despite this, much work continues to assume without formal proof that the inter-annotator agreement gives an upper bound on the achievable machine learning model performance (Boguslav & Cohen, 2017; Richie et al., 2022); the mounting empirical evidence against which is now placed on a solid theoretical footing by the present work.

## 5 CONCLUSIONS

In this paper, we built a theory towards estimating the oracle accuracy of classifiers. Our theory covers *i)* the upper bounds for the average performance of human annotators, *ii)* lower bounds for machine learning models, and *iii)* confidence scores which formally capture the degree of certainty to which we may assert that a model outperforms human annotators. Our theory provides formal guarantees even within the highly practically relevant realistic setting of a finite data sample and no access to an oracle to serve as the ground truth. Our experiments on synthetic classification tasks validate the plausibility of the assumptions on which our theorems are built. Finally, our meta analysis of existing progress succeeded in identifying some existing state-of-the-art models have already achieved superhuman performance compared to the average human annotator.

## BROADER IMPACT

Our approach can identify classification models that outperform typical humans in terms of classification accuracy. Such conclusions influence the understanding of the current stage of research on classification, and therefore potentially impact the strategies and policies of human-computer collaboration and interaction. The questions we may help to answer include the following: *When should we prefer a model's diagnosis over that of a medical professional? In courts of law, should we leave sentencing to an algorithm rather than a Judge?* These questions and many more like them are too important to ignore. Given recent progress in machine learning we believe the work is overdue.

## LIMITATIONS

Yet we caution that estimating a model's oracle accuracy in this way is not *free*. Our approach requires the results from multiple annotators and preferably also the number of annotators should be higher than the number of possible classes in the target classification task. Another potential challenge in applying our analysis is that some of our assumptions may not hold under some specific tasks or settings, e.g., collusion attack by a group of annotators. We recommend those who apply our theory where possible to collect a small amount of 'oracle' annotations, to validate the assumptions in this paper. Our work focus on multi-class classification, which only admits a single answer for each task. A multi-label classification task can be transformed to multiple binary classification tasks before using our theorem.

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

## A   PROOF FOR THEOREMS AND LEMMAS

**Proof of Theorem 1 (Average Performance Upper Bound)**

**Proof** *For $i \neq j$ and $i, j \in \{1, \cdots, K\}$, we have*

$$
\begin{aligned}
\mathbb{P}(\ell_i = \ell_j) =& \mathbb{P}(\ell_i = \ell_j | \ell_j = \ell_\star)\mathbb{P}(\ell_j = \ell_\star) + \\
& \mathbb{P}(\ell_i = \ell_j | \ell_j \neq \ell_\star)\mathbb{P}(\ell_j \neq \ell_\star) \\
\geq& \mathbb{P}(\ell_i = \ell_j | \ell_j = \ell_\star)\mathbb{P}(\ell_j = \ell_\star) \\
=& \mathbb{P}(\ell_i = \ell_\star | \ell_j = \ell_\star)\mathbb{P}(\ell_j = \ell_\star) \\
\geq& \mathbb{P}(\ell_i = \ell_\star)\mathbb{P}(\ell_j = \ell_\star).
\end{aligned}
\tag{23}
$$

*While for $i = j$, we have $\mathbb{P}(\ell_i = \ell_j) = 1$. Therefore,*

$$
\mathbb{P}(\ell_i = \ell_j) \geq \mathbb{P}(\ell_i = \ell_\star)\mathbb{P}(\ell_j = \ell_\star).
\tag{24}
$$

*Then, combining equation 23 and equation 24,*

$$
\mathbb{P}(\ell_{\mathcal{K}} = \ell_\star)^2 = \frac{1}{K^2} \sum_{i=1}^{K} \mathbb{P}(\ell_i = \ell_\star) \sum_{j=1}^{K} \mathbb{P}(\ell_j = \ell_\star)
\tag{25}
$$

$$
\leq \frac{1}{K^2} \sum_{i=1}^{K} \sum_{j=1}^{K} \mathbb{P}(\ell_i = \ell_j)
\tag{26}
$$

$$
\mathbb{P}(\ell_{\mathcal{K}} = \ell_\star) \leq \sqrt{\frac{1}{K^2} \sum_{i=1}^{K} \sum_{j=1}^{K} \mathbb{P}(\ell_i = \ell_j)}.
\tag{27}
$$

**Proof of Theorem 3 (Performance Lower Bound)**

**Proof**

$$
\begin{aligned}
\mathbb{P}(\ell_a = \ell_b) =& \mathbb{P}(\ell_b = \ell_a | \ell_a \neq \ell_\star)\mathbb{P}(\ell_a \neq \ell_\star) + \mathbb{P}(\ell_b = \ell_a | \ell_a = \ell_\star)\mathbb{P}(\ell_a = \ell_\star) \\
\leq& \mathbb{P}(\ell_b = \ell_\star | \ell_a \neq \ell_\star)\mathbb{P}(\ell_a \neq \ell_\star) + \mathbb{P}(\ell_b = \ell_a | \ell_a = \ell_\star)\mathbb{P}(\ell_a = \ell_\star) \\
=& \mathbb{P}(\ell_b = \ell_\star | \ell_a \neq \ell_\star)\mathbb{P}(\ell_a \neq \ell_\star) + \mathbb{P}(\ell_b = \ell_\star | \ell_a = \ell_\star)\mathbb{P}(\ell_a = \ell_\star) \\
=& \mathbb{P}(\ell_b = \ell_\star).
\end{aligned}
\tag{28}
$$

**Proof of Lemma 2 (Convergence of Empirically Approximated Upper Bound)**

**Proof** *By comparing the upper bound and empirical upper bound, we have*

$$
\begin{aligned}
\frac{\mathcal{U}}{\mathcal{U}^{(e)}} =& \sqrt{\frac{K-1}{K} \frac{\sum_{i=1}^{K} \sum_{j=1}^{K} \mathbb{P}(\ell_i = \ell_j)}{\sum_{i=1}^{K} \sum_{\substack{j=1 \\ i \neq j}}^{K} \mathbb{P}(\ell_i = \ell_j)}} \\
=& \sqrt{\frac{K-1}{K}} \sqrt{1 + \frac{\sum_{i=1}^{K} \mathbb{P}(\ell_i = \ell_i)}{\sum_{i=1}^{K} \sum_{\substack{j=1 \\ i \neq j}}^{K} \mathbb{P}(\ell_i = \ell_j)}} \\
=& \sqrt{\frac{K-1}{K}} \sqrt{1 + \frac{K}{\sum_{i=1}^{K} \sum_{\substack{j=1 \\ i \neq j}}^{K} \mathbb{P}(\ell_i = \ell_j)}}.
\end{aligned}
\tag{29}
$$

*For the first factor in equation 29,*

$$
\lim_{K \to +\infty} \sqrt{\frac{K-1}{K}} = 1.
\tag{30}
$$

*For the second factor in equation 29, as both annotators address the same task, the annotator agreement should be better than guessing uniformly at random, i.e. $\mathbb{P}(\ell_i = \ell_j) \geq 1/N_c$, where $N_c$ is*

*the number of categories in the classification task. Then, using a looser constraint $\sum_{j=1, i \neq j}^{K} \mathbb{P}(\ell_i = \ell_j) \geq \frac{K-1}{N_c}$, we have*

$$0 \leq \frac{K}{\sum_{i=1}^{K} \sum_{\substack{j=1 \\ i \neq j}}^{K} \mathbb{P}(\ell_i = \ell_j)} \leq \frac{N_c}{K-1}.$$

*As $\lim_{K \to +\infty} \frac{N_c}{K-1} = 0$,*

$$\lim_{K \to +\infty} \sqrt{1 + \frac{K}{\sum_{i=1}^{K} \sum_{\substack{j=1 \\ i \neq j}}^{K} \mathbb{P}(\ell_i = \ell_j)}} = 1. \tag{31}$$

*Combining equation 30 and equation 31, we have*

$$\lim_{K \to +\infty} \frac{\mathcal{U}}{\mathcal{U}^{(e)}} = 1. \tag{32}$$

*Therefore, the empirically approximated upper bound converges to the theoretical upper bound when $K$ grows larger.*

**Proof of Theorem 5 (Sample Average Performance Upper Bound)**

**Proof** *We apply Theorem 4 with*

$$X_n = \frac{1}{K^2} \sum_{i=1}^{K} \sum_{j=1}^{K} \left[ \ell_i^{(n)} = \ell_j^{(n)} \right], \tag{33}$$

*obtaining $X_n \in [0, 1]$, i.e. $\alpha = 0$, and $\beta = 1$. Let*

$$\mathcal{U}_N \triangleq \sqrt{\frac{1}{K^2} \sum_{i=1}^{K} \sum_{j=1}^{K} \mathbb{P}^{(N)}(\ell_i = \ell_j)}. \tag{34}$$

*Our choice equation 33 of $X_n$ implies $\mathcal{U}_N^2 = \overline{X}$ and $\mathcal{U}^2 = \mathbb{E}[\overline{X}]$, and so by equation 8,*

$$\mathbb{P}\left( \sqrt{t_u + \mathcal{U}_N^2} \leq \mathcal{U} \right) \leq \delta_u. \tag{35}$$

*Rewrite equation 3 as*

$$\mathbb{P}(\ell_{\mathcal{K}} = \ell_\star) \leq \mathcal{U}, \tag{36}$$

*which implies*

$$\mathbb{P}\left( \sqrt{t_u + \mathcal{U}_N^2} \leq \mathbb{P}(\ell_{\mathcal{K}} = \ell_\star) \right) \leq \mathbb{P}\left( \sqrt{t_u + \mathcal{U}_N^2} \leq \mathcal{U} \right). \tag{37}$$

*Combining equation 35 with equation 37 gives the result.*

**Proof of Theorem 6 (Sample Performance Lower Bound)**

**Proof** *We apply Theorem 4 with*

$$X_n = \left[ \ell_a^{(n)} = \ell_b^{(n)} \right], \tag{38}$$

*obtaining $X_n \in [0, 1]$, i.e. $\alpha = 0$, and $\beta = 1$. Let*

$$\mathcal{L}_N \triangleq \mathbb{P}^{(N)}(\ell_a = \ell_b). \tag{39}$$

*Now equation 38 implies $\mathcal{L}_N = \overline{X}$ and $\mathcal{L} = \mathbb{P}(\ell_a = \ell_b) = \mathbb{E}[\overline{X}]$,*

$$\mathbb{P}\left( \mathcal{L}_N - t_l \geq \mathcal{L} \right) \leq \delta_l. \tag{40}$$

*Recall equation 7, $\mathbb{P}(\ell_a = \ell_b) \leq \mathbb{P}(\ell_b = \ell_\star)$, which implies*

$$\mathbb{P}\left( \mathcal{L}_N - t_l \geq \mathbb{P}(\ell_b = \ell_\star) \right) \leq \mathbb{P}\left( \mathcal{L}_N - t_l \geq \mathcal{L} \right). \tag{41}$$

*Combining equation 40 with equation 41 gives the result.*

**Proof of Theorem 7 (Confidence of Out-Performance)**

**Proof** *Recall Theorem 5 and Theorem 6,*

$$\mathbb{P}\left(\sqrt{t_u + \mathcal{U}_N^2} \leq \mathbb{P}(\ell_{\mathcal{K}} = \ell_\star)\right) \leq \delta_u$$

$$\mathbb{P}\left(\mathcal{L}_N - t_l \geq \mathbb{P}(\ell_{\mathcal{M}} = \ell_\star)\right) \leq \delta_l.$$

*Then, we have*

$$\mathbb{P}\left(\mathbb{P}(\ell_{\mathcal{M}} = \ell_\star) - \mathbb{P}(\ell_{\mathcal{K}} = \ell_\star) \geq \tau\right)$$

$$=\mathbb{P}\left(\mathbb{P}(\ell_{\mathcal{M}} = \ell_\star) - \mathbb{P}(\ell_{\mathcal{K}} = \ell_\star) \geq \mathcal{L}_N - t_l - \sqrt{t_u + \mathcal{U}_N^2}\right)$$

$$\geq\mathbb{P}\left(\mathbb{P}(\ell_{\mathcal{M}} = \ell_\star) \geq \mathcal{L}_N - t_l \cap \mathbb{P}(\ell_{\mathcal{K}} = \ell_\star) \leq \sqrt{t_u + \mathcal{U}_N^2}\right)$$

$$\geq 1 - \mathbb{P}\left(\mathbb{P}(\ell_{\mathcal{M}} = \ell_\star) \leq \mathcal{L}_N - t_l)\right) - \mathbb{P}\left(\mathbb{P}(\ell_{\mathcal{K}} = \ell_\star) \geq \sqrt{t_u + \mathcal{U}_N^2}\right)$$

$$\geq 1 - \delta_l - \delta_u. \tag{42}$$

## B  AN EXAMPLE FOR THE ASSUMPTIONS

Here, we provide a running example to show that both assumptions for Theorem 1 and 3 could reasonably hold with no conflict. A common example is demonstrated in Table 4.[7] In this case, all annotators did a decent job (generally more correct than incorrect in all conditions). For the more challenging condition (other annotators fail), the ratio of correct performance is slightly less, see the rows in Table 4a.

|  | $\ell_b = \ell_*$ | $\ell_b = \ell_\times$ |
|---|---|---|
| $\ell_a = \ell_*$ | 0.8 | 0.2 |
| $\ell_a = \ell_\times$ | 0.6 | 0.4 |

(a) $\mathbb{P}(\ell_b = \ell_? | \ell_a = \ell_?)$

| | |
|---|---|
| $\mathbb{P}(\ell_a = \ell_*)$ | 0.7 |
| $\mathbb{P}(\ell_a = \ell_\times)$ | 0.3 |

(b) $\mathbb{P}(\ell_a = \ell_?)$

Table 4: An example for the assumptions.

For assumption 1, all possible inequations hold:

- $0.74 = 0.56 + 0.18 = \mathbb{P}(\ell_b = \ell_*) \leq \mathbb{P}(\ell_b = \ell_* | \ell_a = \ell_*) = 0.80$
- $0.26 = 0.14 + 0.12 = \mathbb{P}(\ell_b = \ell_\times) \leq \mathbb{P}(\ell_b = \ell_\times | \ell_a = \ell_\times) = 0.30$
- $0.70 = 0.56 + 0.14 = \mathbb{P}(\ell_a = \ell_*) \leq \mathbb{P}(\ell_a = \ell_* | \ell_b = \ell_*) = 0.56/0.74 = 0.757$
- $0.30 = 0.18 + 0.12 = \mathbb{P}(\ell_a = \ell_\times) \leq \mathbb{P}(\ell_a = \ell_\times | \ell_b = \ell_\times) = 0.12/0.26 = 0.462$

For assumption 2, all possible inequations hold:

- $0.6 = \mathbb{P}(\ell_b = \ell_* | \ell_a = \ell_\times) \geq \mathbb{P}(\ell_b = \ell_\times | \ell_a = \ell_\times) = 0.4$
- $0.538 = 0.14/0.26 = \mathbb{P}(\ell_a = \ell_* | \ell_b = \ell_\times) \geq \mathbb{P}(\ell_a = \ell_\times | \ell_b = \ell_\times) = 0.462$

Note that, if $b$ should be a decent ML model or a rational annotator who works better than random guessing, *i.e.*, $0.5 = \mathbb{P}(\ell_b = \ell_* | \ell_a = \ell_\times) = \mathbb{P}(\ell_b = \ell_\times | \ell_a = \ell_\times) = 0.5$.

---

[7]Binary classification is discussed for simplicity.

## C    DETAILS FOR NLP DATASETS

Table 5: Statistics of SST and SNLI: the number of test samples, number of classes, and the number of annotators for each sample. Note that annotators are sampled from a large and diverse pool.

| Dataset | #Test | #Class | #Annot. |
|---|---|---|---|
| SST-2 (Socher et al., 2013) | 1,821 | 2 | 3 |
| SST-5 (Socher et al., 2013) | 2,210 | 5 | 3 |
| SNLI (Bowman et al., 2015) | 10,000 | 3 | 5 |

## D    DETAILS FOR HUMAN ANNOTATION

We crowd source the annotations via the *Amazon Mechanical Turk*. The annotation interfaces with instructions for color classification and shape classification are illustrated in Figure 4. Each example is annotated by $K = 10$ different annotators. For quality control, we *i)* offer our tasks only to experienced annotators with 100 or more approved HITs; *ii)* automatically reject answers from annotators who have selected an invalid option 'None of the above'.

We demonstrate the inter-annotator agreement (Cohen's Kappa, Fleiss' Kappa and Krippendorff's Alpha) of collected annotations on Color and Shape, in Table 6. Note that Cohen's Kappa compares only two annotators. We calculate the mean of Cohen's Kappa scores between all $K(K - 1)/2$ different pairs of annotators. The results show that our collected human annotation datasets cover the cases for both strongly (Color) and weakly (Shape) correlated human annotations.

Table 6: Inter-annotator agreements on classification tasks, Color and Shape.

| Task | Cohen's Kappa (mean) | Fleiss' Kappa | Krippendorff's alpha |
|---|---|---|---|
| Color | 0.6040 | 0.6036 | 0.5819 |
| Shape | 0.2386 | 0.2372 | 0.2330 |

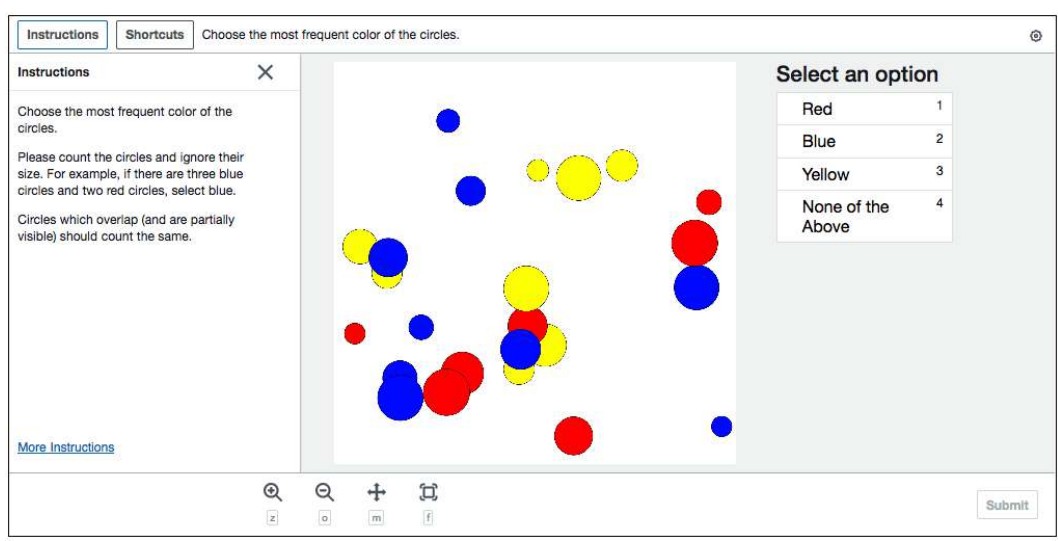

(a) Color Classification

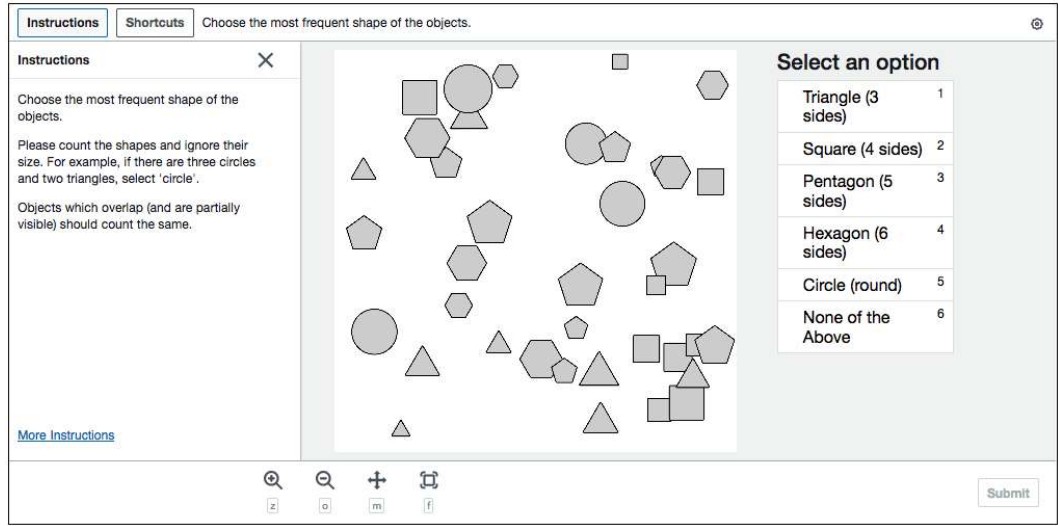

(b) Shape Classification

Figure 4: Human annotation interface for the Color Classification and Shape Classification tasks.

