# OpenReview forum: "Humanly Certifying Superhuman Classifiers"
_ICLR.cc/2023/Conference — ICLR 2023 notable top 25%_

### Official Review · Reviewer_xVdj · 2022-10-25

**Confidence:** 4
**Correctness:** 3
**Technical Novelty And Significance:** 2
**Empirical Novelty And Significance:** 2
**Recommendation:** 5

**Clarity, Quality, Novelty And Reproducibility:**

The paper brings an element of novelty, but the writing is not clear. Many clarifications are needed. In addition, it is not clear how the work builds upon prior literature that does not assume that all annotators are equally skilled. More explanations behind this emphasis on whether the model outperforms the human "average" annotator are needed.

**Strength And Weaknesses:**

The work is interesting and I like the spirit of the paper. However, it was difficult for me to get through it mainly because of the notation used. I describe below some of my concerns.

First, I don't fully understand why the authors focus on finding bounds for the accuracy of the ``average'' annotator ($l_K$) rather than of some other statistics of the distribution of crowdsourced labels, e.g., its mode. In practice, $P(l_K=l_*)$ is not of much interest. It would be way more interesting to study whether the wisdom of the crowds beats the model or vice-versa. One issue is that the proposed framework seems to assume that all workers have the same ability, which is certainly not the case. Most models used in crowdsourcing do not make such an assumption, which represents a limitation of this framework (and these annotators should not be considered at all!). If the authors have some specific use cases for the model that they propose, they should describe them (decision-making with experts could be one).

The notation is fairly confusing in my opinion. For example, 2.1 first introduces $l_i$ and then reintroduces $l_a$, which in 2.1 it is presented as the label assigned by the annotator while in 2.3 as the label predicted by the model. It should be mentioned that i and j are in ${1,\dots, K}$. It would be useful to clarify that $P(l_i =l_j)$ can vary with i,j. The theoretical upper bound $U^{(t)}$ should be defined already in (3). Hoeffding's inequality is a standard inequality, thus there is no need to report its statement. It is unclear what the sample size $N$ refers in page 4: While it should indicate the number of data points (probably tasks? as explained in page 2), in equation (9) it seems that for each data point the annotations of two fixed annotators are considered for each data point (task?). Why?

In section 2.5, bullet point 3: What is the "margin"? It also seems that the model performance is assessed against some ``aggregated'' version of the human annotations; one should take into account the likelihood that the majority vote or some other sort of aggregate version may not correspond to the real ground truth.

Other clarifications:
* What does "calibrate this overestimation" mean?

* For the case where the model outperforms humans only in certain regions of the feature space, how is this handled by the proposed framework? This seems to be an important consideration.

* Assumption in (6) probably needs some discussion because it seems possible that if the tasks admits several plausible answers

* Confidence bands in the experiments plots should be reported.

* Details on IRB approval need to be reported, including how much workers were paid in terms of hourly wage.


**Summary Of The Paper:**

This work investigates whether it is possible to assess when statistical models outperform humans in crowdsourcing tasks. In this context, the main issue is that ground truth is not available and human annotations are generally used as proxies. The authors find deterministic and probability bounds for the accuracy of what they call the "average" human annotator. They then propose a method to find whether the model outperforms humans and show its applicability in a series of experiments.

**Summary Of The Review:**

I have raised several concerns that I hope the authors will address in their rebuttal.

---

> ### Author Response · Authors · 2022-11-10
> **Response to Reviewer xVdj (Part 2)**
>
> **Q9:** What does "calibrate this overestimation" mean?
>
> **A9:** Because $P(\ell_i=\ell_j)=1$ when we are comparing the same annotator $i=j$, we consider the inclusion of this in the estimation of the average inter-annotator agreement as a giving rise to an “overestimation”. Usually, we are interested in the rate of agreement of different annotators, namely $i \neq j$, where $\mathbb{P}(\ell_i=\ell_j) \leq 1$. For the proof of Theorem 1 in Appendix A, we did not manage to avoid including the case of $i=j$. Intuitively however, comparing annotators’ behaviors with themselves is inappropriate and this term always diminishes our estimation of the human upper bound. An empirical approach to avoid it is provided in Definition 1, and fortunately we manage to prove that it will converge to the theoretical upper bound when we take the limit of large $K$.
>
> **Q10:**  For the case where the model outperforms humans only in certain regions of the feature space, how is this handled by the proposed framework? This seems to be an important consideration.
>
> **A10:** We only work with the predicted labels ; our method, like the often reported accuracy of the model and inter-annotator agreement, is agnostic to the actual underlying inputs in the supervised learning problem that give rise to the labels. Extending our work the case you describe would be technically interesting and challenging, but perhaps less relevant in terms of comparing different published results, which usually only report the quantities described. Moreover, your idea of considering “certain regions” would require complex assumptions about similarity in feature space.
>
> **Q11:** Assumption in (6) probably needs some discussion because it seems possible that if the tasks admit several plausible answers.
>
> **A11:** We have provided a running example for explaining both assumptions (in Equation 2 and Equation 6) to reviewer gbkc. We hope it also addresses your concerns. For the tasks, we consider multi-class classification tasks, which only admit a single answer for each. A multi-label classification task can be transformed to multiple binary classification tasks before applying our theorem.  We have added this explanation to the section of limitations. We are happy to further explain any other concerns which you may have.
>
> **Q12:** Details on IRB approval need to be reported, including how much workers were paid in terms of hourly wage.
>
> **A12:** The averaged wage is about 12 USD / hour, excluding the rejected annotations.
>
> We hope our responses and revision have addressed your questions and consolidated our theory.

---

> ### Author Response · Authors · 2022-11-10
> **Response to Reviewer xVdj (Part 1)**
>
> Thank you for your comments and suggestions. We hope the following clarifications help to address your concerns.
>
> **Q1:** I don't fully understand why the authors focus on finding bounds for the accuracy of the ``average'' annotator
>
> **A1:** We would like to highlight that we are interested in two bounds: 1) the upper bound for the averaged annotator performance in Theorem 1, and ii) *the lower bound for the model performance* in Theorem 3. The outperformance can only be certified when the second one is higher than the first one with sufficient margin of confidence. Some intriguing propositions are that 1) our lower bound is well connected with the existing widely reported inter-annotator agreement (square root of the rate of agreement between all annotators); and 2) our upper bound could be specified by the also exceptionally common practice of accuracy calculation (accuracy on aggregated human annotations provided by the benchmark datasets). Many authors are aware of these two practices, but connecting them in a novel theoretical framework under the view of *oracle accuracy* was challenging, and we hope, intriguing and useful. Moreover, our theory and proposed algorithm provides a probabilistic confidence guarantee of the outperformance which is easy to evaluate.
>
> **Q2:**  It would be way more interesting to study whether the wisdom of the crowds beats the model or vice-versa.
>
> **A2:** Actually, we are working towards this interesting direction, by composing an upper bound $\mathcal U$ for the crowds and a lower bound $\mathcal L$ for the model. If we observe $\mathcal L \geq \mathcal U$, we may come up with some interesting claims such as that a model beats some crowds on a specific task, etc.
>
> **Q3:** One issue is that the proposed framework seems to assume that all workers have the same ability, which is certainly not the case.
>
> **A3:** We do not have such an assumption. The performance of the annotators with different abilities could be averaged and it could be considered as the *‘wisdom or ability’ of a group*. For example, teachers average the scores of all students in a class to estimate the performance of this group.
>
>
> **Q4:** The notation is fairly confusing in my opinion. For example, 2.1, first introduces $l_i$ and then reintroduces $l_a$,
>
> **A4:** Sorry for the confusion. $i$ and $j$ are used to differentiate two annotators in a group with $K$ people (mostly for upper bounding). $\ell_a$ and $\ell_b$ are for the lower bounds.  Although $a$ and $b$ could be both human and model, we prefer to use $a$ for the (aggregated) human ($\ell_a=\ell_{\mathcal G}$) and $b$ for the machine learning model ($\ell_b=\ell_{\mathcal M}$). We have modified Section 2 for better explanation.
>
>
> **Q5:** Hoeffding's inequality is a standard inequality, thus there is no need to report its statement.
>
> **A5:** Hoeffding’e inequation is a grounding lemma to prove Theorem 5 and Theorem 6. These two theorems are important to drive the algorithms for outperformance confidence scores. As there could be multiple versions of Hoeffding theorem, we claim it in our paper for better reference and explanation. If you prefer, we can move it to the appendix.
>
> **Q6:** It is unclear what the sample size N refers to in page 4. While it should indicate the number of data points (probably tasks? as explained in page 2)
>
> **A6:** It is the number of data samples. Thanks for this comment, we have clarified this in our revision.
>
> **Q7:** In equation (9) it seems that for each data point the annotations of two fixed annotators are considered for each data point (task?). Why?
>
> **A7:** This is one pair of the $i$-th and $j$-th annotators. We consider all pairs between $K$ annotators for each data sample, see the summation on the right-hand side of Equation (11).
>
> **Q8:** In section 2.5, bullet point 3: What is the "margin"?
>
> **A8:** The margin is the difference between the upper and lower bounds. Because we are interested in the outperformance of the model (lower bounded by $\mathcal L$) in comparison with the averaged human (upper bounded by $\mathcal U$), it is intuitive that if we wish to claim with high confidence that the model out-performs, then it is necessary but not sufficient that at least $\mathcal L_N - \mathcal U_N \geq 0$.  A larger margin, say a high positive value, is of greater interest to the verification as it gives higher confidence of outperformance, and the more precise approach we give further accounts for the finite sample size. We have further clarified this in our revision.

---

### Official Review · Reviewer_ofWq · 2022-10-27

**Confidence:** 3
**Correctness:** 4
**Technical Novelty And Significance:** 4
**Empirical Novelty And Significance:** 4
**Recommendation:** 8

**Clarity, Quality, Novelty And Reproducibility:**

The paper is well-written. The experiments are clearly presented and easy to understand but the theory can be confusing with changing notations (and some not well defined). The paper offers a new perspective on super-human performance and its novelty lies in the same. Experiments should be reproducible without a ton of effort, though I haven't tried reproducing them (and the authors haven't submitted their code as a supplementary attachment).

**Strength And Weaknesses:**

- This paper is well motivated and addresses an important problem in machine learning. It would be widely of interest to the community.
- The paper is mostly well-written and provides a detailed description of the proposed theory and algorithm.
- The toy setup along with empirical results on real world datasets make the case for the validity of the theory. The results appear to be promising, with some existing models already achieving superhuman performance with high probability.
- The theoretical setup can be confusing at times. I recommend the authors do a pen and paper read, go line by line and ask themselves, what does this mean?
 - As an example, it's not clear why certain notation has been redefined (l_{i} in first sentence of Section 2.1 and then l_{a} for the same thing in the next sentence).
 - There are also several typos in the paper ("inter-annotator aggreement", "empirical works has", etc.)
- One question I would encourage the authors to think about: how does this work in cases where the labels may not be as objective as in sentiment or NLI. Take hate speech classification for example. What one person may think is acceptable speech, another might find offensive. Doesn't mean that either is wrong, all it means is that their labels rely on their lived experiences. How will your method work in those cases? And if it won't, why? What are the limitations of this approach?

Happy to update my score post revision.

**Summary Of The Paper:**

The paper discusses the idea that oracle labels for classification tasks may be unobservable, and as a result, researchers often resort to two heuristics: human predictions or aggregated human annotations are effectively treated as ground truth to approximate the oracle, and the inter-annotator agreement is taken as the best possible machine learning model performance. The authors argue that this approach has several disadvantages, including the challenges of quality control for human annotation and the fact that current evaluation paradigms focus on evaluating the performance of models, but not the oracle accuracy of humans. They propose a theory for estimating the oracle accuracy on classification tasks which formalizes that machine learning classification models may outperform the humans who provide them with training supervision. The theory includes upper bounds for the averaged oracle accuracy of the annotators, lower bounds for the oracle accuracy of the model, and finite sample analysis for both bounds and their margin which represents the model’s outperformance. Based on this, they propose a method to detect competitive models and to report confidence scores, which formally bound the probability that a given model outperforms the average human annotator. Empirically, it is observed that some existing models for sentiment classification and NLI have already achieved superhuman performance with high probability.

**Summary Of The Review:**

The paper is well motivated, well-written and provides a detailed exploration of an important issue around classification tasks---to estimate bounds for the oracle performance and determine whether a model has outperformed human annotators. The paper can be confusing at times but it is something that I believe can be addressed in camera ready. I would also like to see more discussion around limitations of this work. The proposed algorithm could be useful in practice, particularly in scenarios where it is difficult to obtain accurate ground truth labels.

---

> ### Author Response · Authors · 2022-11-10
> **Response to Reviewer ofWq**
>
> Thank you for your appreciation and insightful feedback.
>
> **Q1:** The theoretical setup can be confusing at times. I recommend the authors do a pen and paper read, go line by line and ask themselves, what does this mean?
>
> **A1:** We have modified Sec 2.1, 2.3, 2.4. Two new sections are added for broader impact and limitations of our work. We have also given small seminars in multiple research groups and we have optimized our notations and proofs based on the feedback we have received. We hope the modifications and clarification have addressed your concerns. We welcome further questions on specific parts of the paper.
>
> **Q2:** Why certain notation has been redefined (l_{i} in first sentence of Section 2.1 and then l_{a} for the same thing in the next sentence).
>
> **A2:** A brief answer is that they are for different parts of analysis. $i$ and $j$ are used to differentiate two annotators in a group of $K$ people (used for describing the upper bounds). Although $a$ and $b$ could be both human and model, we prefer to use $a$ for the (aggregated) human ($\ell_a=\ell_{\mathcal G}$) and $b$ for the models ($\ell_b=\ell_{\mathcal M}$), when describing the lower bounds.
>
> **Q3:** There are also several typos in the paper.
>
> **A3:** Thank you for these suggestions. We have modified the paper accordingly.
>
> **Q4:** How does this work in cases where the labels may not be as objective as in sentiment or NLI. How will your method work in those cases? And if it won't, why? What are the limitations of this approach?
>
> **A4:** This question is more of a philosophical one about what makes a task "truly subjective or objective". For example, in the task of predicting user ratings given the profile of a user and items, the rating itself is subjective to the user's own preference and value. But in the context of modelling, we understand that there might be an oracle probability distribution that best describes such relations. It is thus also reasonable to imagine a "superhuman" model which outperforms humans in predicting these "subjective" ratings. For limitations, we have added a section to unveil those we are aware of.
>
> **[Reproducibility]:** Experiments should be reproducible without a ton of effort, though I haven't tried reproducing them (and the authors haven't submitted their code as a supplementary attachment).
>
> **A:** Thank you for your interest and the suggestion. We will publish our (relatively simple) analytical code and data to the public upon acceptance.
>
> We hope our responses and revision have addressed your questions and consolidated our theory.

---

### Official Review · Reviewer_gbkc · 2022-10-30

**Confidence:** 4
**Correctness:** 3
**Technical Novelty And Significance:** 4
**Empirical Novelty And Significance:** 2
**Recommendation:** 6

**Clarity, Quality, Novelty And Reproducibility:**

The paper is a bit difficult to read and requires multiple passes. While the main message and methodology is clear, the clarity is hindered by the present structure of the paper.   I would have expected a more structured discussion of the nature of human annotation, central assumptions about the population of annotators and a definition of what it means to be superhuman and connect this to the specific derivation of the bounds.  The disclaimer about the application of the methodology and the need for careful validation could be articulated early on so that the reader is better positioned to clearly view what is claimed.

The paper is original and is of good quality. I like the theme and the overall approach.  Grounding the claims with clarity will help strengthen the paper.  There is no mention in the paper about providing the sources for reproducing the experiments.

**Strength And Weaknesses:**

Strengths:
Addresses a very important problem of subjectivity and randomness among human annotators and how one can verify whether a machine learning model comes close to an ideal (unknown) oracle performance.  The theory, proofs and experiments adequately illustrate the potential utility of the methodology.

Weaknesses:
My concern about the paper has to do with the definition of what constitutes superhuman and the comparison of the ML model against the bound on average human performance.  The source of variability in annotations is complex and involves many variables - the availability and cost of getting best annotations from human experts, the context sensitivity of tasks and of human expert performance variability depending on their experience, etc. Thus, I would expect that we ought to be deriving bounds for the performance achievable for the best picked human annotator for a given context and then contrasting the ML model performance against the best subset of humans for a given instance.  I am not absolutely clear about assumption (2) discussed in my summary above, as this in some sense seems contradicting to assumption (1) that the human annotators are correlated. If the human annotation results are indeed correlated, how is it that given that one annotation result that is incorrect another human annotator will have a higher likelihood of being correct to the oracle rather than being wrong. There is a mention in the paper that random guessing will do and this is a weak assumption. Can you elaborate on this?

**Summary Of The Paper:**

The paper presents a theory to address the problem of evaluating the performance of both humans and
machine learning models with respect to an oracle which is unobserved. The theory hinges on two assumptions: 1) human annotators are positively correlated and 2) the fact that a particular annotator wrongly labels an instance does not preclude the chance of the other annotator to be predictive and be correct.  These assumptions are shown to lead to an upper bound for the averaged performance across human annotators and a lower bound for the machine learning model performance.  A finite sample analysis is then provided to derive a practical estimator for these bounds and an algorithm for evaluating whether a machine learning model is super-human. The assumptions are validated on toy experiments with known oracles and the utility of the theory is illustrated by meta-analysis in sentiment classification and natural language inference. Based on the paper's definition of what constitutes superhuman performance, it is shown that current ML models may already by superhuman in these tasks.

**Summary Of The Review:**

Overall I like the paper and it is an important contribution.  There is enough in the paper to recommend acceptance. Revision of the paper to address clarity of the claims and discussion of limitations will strengthen the paper.

---

> ### Author Response · Authors · 2022-11-10
> **Response to Reviewer gbkc**
>
>
> Thank you for your appreciation and insightful feedback.
>
> **Q1:** The source of variability in annotations is complex and involves many variables. I would expect that we ought to be deriving bounds for the performance achievable for the best picked human annotator for a given context and then contrasting the ML model performance against the best subset of humans for a given instance.
>
> **A1:** Philosophically, certifying the outperformance against a single best annotator using only the information about that annotator seems very challenging. Presumably, stronger assumptions would be required to make progress. In practice, certifying the outperformance against a *‘best’ subgroup* is plausible. We can collect multiple (sub)groups of annotators, then calculate multiple upper bounds and lower bounds. If one of the lower bounds is higher than all the upper bounds, including the highest one, and with large confidence scores, then we can claim that the model outperforms the ‘best’ subgroup.
>
> **Q2:** If the human annotation results are indeed correlated, how is it that given that one annotation result that is incorrect another human annotator will have a higher likelihood of being correct to the oracle rather than being wrong.
>
> **A2:** This is an excellent point to raise, and we will add words to preempt this question in the mind of future readers. There is no conflict between our two assumptions, indeed:
>
> 1) The two assumptions are conducted on different variables. The 1st assumption is on pairs of annotators, $i, j \in \[1..K\]$. The 2nd assumption is on $a$ (which is preferred to be aggregated annotators $\mathcal G$) and $b$ (which is in our case typically an ML model $\mathcal M$). (also see the explanation after Equation (7)).
>
> 2) The two assumptions can however be shown not to conflict, even on the same pair of random variables. Here we provide a running example with explanation. Binary classification is discussed for simplicity.
>
> | $\mathbb P(b=\ell_? \| a=\ell_?)$| $b=\ell_*$ | $b=\ell_\times$ |
> | :--- | :---: | :---:|
> |$a=\ell_*$ | 0.8 | 0.2 |
> |$a=\ell_\times$ | 0.6 | 0.4 |
>
> |$\mathbb P(a=\ell_*)$|$\mathbb P(a=\ell_\times)$|
>  | :---: | :---:|
> |0.7|0.3|
>
> This example is quite common in practice, as all annotators did a decent job (generally more correct than incorrect in all conditions). For the more challenging condition (other annotators fail), the ratio of correct performance is slightly less.
>
> For assumption 1,
> -  $0.74=0.56+0.18=\mathbb P(b=\ell_*) \leq \mathbb P(b=\ell_* \| a=\ell_*) = 0.80$
> -  $0.26=0.14+0.12=\mathbb P(b=\ell_\times) \leq \mathbb P(b=\ell_\times \| a=\ell_\times) = 0.30$
> -  $0.70=0.56+0.14=\mathbb P(a=\ell_*) \leq \mathbb P(a=\ell_* \| b=\ell_*) = 0.56/0.74=0.757$
> -  $0.30=0.18+0.12=\mathbb P(a=\ell_\times) \leq \mathbb P(a=\ell_\times \| b=\ell_\times) = 0.12/0.26=0.462$
>
> For assumption 2,
> - $0.6=\mathbb P(b=\ell_* \| a=\ell_\times) \geq \mathbb P(b=\ell_\times \| a=\ell_\times)=0.4$
> - $0.538=0.14/0.26=\mathbb P(a=\ell_* \| b=\ell_\times) \geq \mathbb P(a=\ell_\times \| b=\ell_\times)=0.462$
>
> Note that, if $b$ should be a decent ML model or a rational annotator who works better than random guessing, i.e. $0.5=\mathbb P(b=\ell_* \| a=\ell_\times) = \mathbb P(b=\ell_\times \| a=\ell_\times)=0.5$.
>
> Moreover, our Theorem 3 covers multi-class settings. $\ell_{\times}$ indicates any of the *single* incorrect labels. Current notation is the most effective way to correctly express our theory to the best of our knowledge. We are open to any further advice on improving the clearance of our work.
>
> **[Clarity]:** The disclaimer about the application of the methodology and the need for careful validation could be articulated early on so that the reader is better positioned to clearly view what is claimed.
>
> **A:** Thanks for raising this, we have added a section which discusses the limitations of our work.
>
> **[Reproducibility]:** Providing the sources for reproducing the experiments.
>
> **A:** For the synthetic experiments, we will publish the code and data to the public, and we note that it is relatively simple to implement, anyway.
> For the meta-analyses in sentiment classification and natural language inference, we mostly reuse reported accuracies (on aggregated human annotations) from existing papers that claimed to be the then state-of-the-art. That there is no requirement to re-bake the existing reported results is one of the elegant characteristics of our work, making our theory easy to implement and base comparisons on.
>
> We hope our responses and revision have addressed your questions and consolidated our theory, thank you !

---

> > ### Comment · Reviewer_gbkc · 2022-11-14
> > **Clarification addresses my concerns**
> >
> > Thanks for the clarifications. It addresses my concerns sufficiently.

---

> > > ### Author Response · Authors · 2022-11-14
> > > **Thanks to Reviewer gbkc**
> > >
> > > We would like to appreciate your quick response and invaluable feedback.

---

### Official Review · Reviewer_F2Uj · 2022-11-05

**Confidence:** 3
**Correctness:** 4
**Technical Novelty And Significance:** 4
**Empirical Novelty And Significance:** Not applicable
**Recommendation:** 8

**Clarity, Quality, Novelty And Reproducibility:**

I have a few concerns about the clarity and quality of this paper, enumerated below.

**1. Unclear random vs. deterministic quantities in the problem statement**

I think the distinction between what is random and what is deterministic in the problem statement could be clearer.

Since $\mathbb{P}(l_i = l_*)$ is defined as "the ratio of matched labels" in Section 2.1, I believe that the dataset of $N$ points is fixed and not a random sample. Hence, $l_*$ and each $l_i$ is deterministic, and $\mathbb{P}(l_i = l_*)$ is a deterministic ratio. $l_\mathcal{K}$, in contrast, is random. Hence, $\mathbb{P}(l_\mathcal{K} = l_*)$ is random. The usage of $\mathbb{P}$ for both random and deterministic quantities makes following the problem statement a bit difficult.

Later in Theorem 1, $\mathbb{P}(l_i = l_j)$ is treated like a probabilistic quantity, which suggests that $\mathbb{P}(l_i = l_j)$ is the probability that $i$ and $j$ agree on all $N$ labels for a randomly sampled dataset of $N$ points. However, this contradicts Section 2.1 which defines $\mathbb{P}(l_i = l_*)$ as "the ratio of matched labels".

Finally in Theorem 5, $\mathbb{P}^{(N)}(l_i = l_j)$ is defined as the empirical fraction of $N$ data points where $i$ and $j$ agree on the label, which suggests that $\mathbb{P}(l_i = l_*)$ is the "the ratio of matched labels" assuming $N \rightarrow \infty$.

It would help if the notation made the appropriate interpretation of each probability unambiguous.

**2. Unclear meaning of "$\mathbb{P}(l_i = l_j)$ is overestimated as 1"**

Based on Section 2.1, $l_i$ is given for $i=1,\dots,K$. Hence, $\mathbb{P}(l_i = l_j) = 1$ when $i=j$, and $\mathbb{P}(l_i = l_j)$ is a deterministic quantity (the fraction of labels over $N$ data points on which humans $i$ and $j$ agree).

What does it mean to "estimate" or "overestimate" $\mathbb{P}(l_i = l_j)$?

**3. Clarity and restrictiveness of the assumption in Lemma 2**

Lemma 2 assumes that $\mathbb{P}(l_i = l_j) \geq 1/N_c$. Should this be for every pair of humans $i$ and $j$?

I am also unclear on how to interpret this assumption. One interpretation is, assuming the dataset is a random sample, that $\mathbb{P}(l_i = l_j$ is the probability that $i$ and $j$ agree on all labels in the dataset. This will likely be pretty low, so the assumption is unlikely to hold in practice (given that $N_c$ is usually not very large). Another interpretation is that $\mathbb{P}(l_i = l_j)$ is the fraction of labels on with $i$ and $j$ agree on a fixed dataset, in which case this assumption is not restrictive (annotator agreement rates are typically upwards of 70% in practice).

**4. How are the bounds in Figure 3 calculated for the case of just one annotator?"

The formula for the lower bound does not make this clear.

**5. Possible conflict between the assumptions of Theorem 1 and 3**

Theorem 1 assumes that the human annotators are positively correlated. Theorem 3 assumes that even when the aggregated human annotation is incorrect, it is possible for the machine learning model to be predict the label correctly. Given that the machine learning model is trained to mimic human annotations, and that these annotations are correlated, it seems that these 2 assumptions are in conflict.

Considering an extreme case (eg. humans are completely incorrect and strongly correlated, completely incorrect and weakly correlated, etc.) may help illuminate these assumptions better, and help evaluate whether they are applicable to a specific setting in practice. I believe (but am not sure), taken together, the proposed theorems rely on the human annotations being reasonably accurate and reasonably correlated (but not too correlated).

**6. Minor Grammatical/Spelling Errors:**

"Within this setting provide..."
"along some labels"
"for all of a pairs"
"affects their decisions, and etc."
"we introduce another assumption equation"
"in that as even"

**Strength And Weaknesses:**

This paper has several strengths.

It considers an important problem: how can we theoretically guarantee that a given model exceeds human annotation performance? More importantly, it enables qualifying claims of superhuman performance by quantifying whether the observed/reported superhuman performance is statistically significant for a given number of annotators with a specific inter-annotator agreement.

The paper approaches this problem in a principled manner, by deriving upper bounds on human performance and lower bounds on model performance (without needing access to ground truth labels). The paper also derives finite sample variants of these bounds that are practically useful, and derives confidence intervals on the difference between these two bounds. All assumptions are clearly stated.

The paper concludes with an empirical evaluation of the proposed theory, which covers several important aspects of the theory such as the validity of the assumptions, whether the bounds are valid empirically, and how they vary as the number of annotators increases.

**Summary Of The Paper:**

This paper proposes an approach to certify whether a given machine learning model achieves super-human performance when the dataset labels are (possibly erroneous) human annotations and not (unobserved) ground-truth labels. The proposed approached relies on proving the following results (given $K$ human annotators and infinite data):

   1. (Theorem 1) The probability that a randomly selected human annotator labels all samples correctly is bounded above by the root mean squared inter-annotator agreement.

   2. (Theorem 3) The probability that the machine learning model predicts the labels of all samples correctly is bounded below by the probability that the machine learning model predicts labels that match the aggregated human-annotated labels (aggregated via majority voting, for example).

Note that the lower bound in Theorem 3 is basically the machine learning model accuracy with respect to the human annotations, as is traditionally reported. The paper also proves variants of the theorems above for finite data samples.

The theorems above rely on the following assumptions:

   1. Human annotators are not independent; their annotations are "positively correlated".
   2. Even if the aggregated human annotation is incorrect, the machine learning model is more likely to predict the correct label than the incorrect human annotation.

The proposed approach essentially compares the upper bound for human annotators (Theorem 1) with the lower bound for the machine learning model (Theorem 3). The paper also provides a way to construct confidence intervals on the difference between these two bounds. The paper concludes with an empirical evaluation of the proposed theory.

**Summary Of The Review:**

This paper considers the important problem of certifying whether an ML model achieves superhuman performance, and provides a principled approach to doing so. The proposed approach is clear and transparent in its assumptions. However, some of the notation makes it difficult to follow the paper, and the restrictiveness of the assumptions in practice is insufficiently explored. While both these drawbacks are significant, I believe they are addressable.

---

> ### Author Response · Authors · 2022-11-10
> **Response to Reviewer F2Uj**
>
>
> Thank you for your appreciation and an excellent summarization of our work.
>
> **Q1:** Unclear random vs. deterministic quantities in the problem statement
>
> **A1:** The expressions such as $\mathbb {P}(l_i=l_j)$ are unknown true probabilities while the  $\mathbb {P}^{(N)}(l_i=l_j)$ is the empirical counterpart given $N$ observations, and the former is the limit $N\rightarrow \infty$ of the latter, as you correctly state. While the former is a probability, this may be treated probabilistically in a Bayesian way (such as by placing a prior on a Bermoulli probability), or, as in our case, with concentration type inequalities that are based on (probabilistically!) bounding the deviation between the two (unknown true probability and empirically observed empirical probability). We agree that our explanation can be made more clear and friendly to the reader, and we have added additional explanations to Sec 2.4.
>
> **Q2:** Unclear meaning of $\mathbb{P}(l_i=l_j)$ is overestimated as 1.
>
>
> **A3:** Thanks for drawing attention to it. We should say instead that the average inter-annotator agreement will be over-estimated by including the self comparison terms $\mathbb{P}(l_i=l_j)$ when $i=j$, which is always equal to one. This leads to an intuitive idea to avoid the inclusion of these terms, which we demonstrate is valid in the limit $ K \rightarrow \infty$.
>
> **Q3:** Clarity and restrictiveness of the assumption in Lemma 2.
>
> **A3:** If the annotation is based on random guesses, the probability of matched labels from two annotators is $\mathbb{P} (\ell_i=\ell_j)=\frac{1}{N_c}$. Given the annotators are decently addressing the same task, their agreement should be better than guessing uniformly at random. We have reviewed this lemma, and it is possible to prove it using a looser constraint based on the averaged match between K-1 annotators, $\sum_{\substack{j=1,i\neq j}}^{K}\mathbb{P}(\ell_i = \ell_j) \geq \frac{K-1}{N_c}$. Please check our revision to the statement (Lemma 2) and the corresponding proof in Appendix A (Proof of Lemma 2).
>
> **Q4:** How are the bounds in Figure 3 calculated for the case of just one annotator?
>
> **A4:** For upper bounds $\mathcal U$, our lines (brown and red) start from two annotators. For lower bounds $\mathcal L$, ‘the aggregation of one annotator’ is based on the single annotator. We have made it more specific in the caption of Figure 3.
>
> **Q5:** Possible conflict between the assumptions of Theorem 1 and 3
>
> **A5:**
>
> This is an excellent point to raise, and indeed our particular choice of assumptions is a key contribution, and we believe that they are reasonable. We provide several lines of reasoning to support this:
>
> 1) To ‘Theorem 3 assumes that even when the aggregated human annotation is incorrect, it is possible for the machine learning model to predict the label correctly.’, we would like to clarify that we assume that $\ell_b$ is reasonably predictive even on those instances that $a$ gets wrong’. ‘Reasonably predictive’ does not imply superb correctness, as it will depend on the performance from both correctly and incorrectly labeled examples. For example, a random guess can satisfy this by giving equal probability to the oracle and the other labels on all incorrect labels, but its actual performance on the whole dataset remains unknown.
>
> 2) To the case that ‘humans are completely incorrect and strongly correlated’, namely annotators always choose the same wrong answer, this would mean that the collected dataset is unreliable for the task. Such a problematic case must be (and typicaly would be) avoided the data collection protocol. To ‘completely incorrect and weakly correlated’, in this case the annotators are not good at this task and the upper bound of their averaged accuracy should be low, therefore, certifying a model outperforms this (weak) group is possible.
>
>
> 3) We have provided a running example for binary classification to Reviewer gbkc. The key assumptions are also validated on human data with known oracles in Sec 3.2 RQ2. Please have a check if you still have concern with the assumptions.
>
> **Q6:** Minor Grammatical/Spelling Errors.
>
> **A6:** Thanks for the suggestions. We have incorporated them in our revision.
>
> We hope our responses and revision have addressed your questions and consolidated our theory.

---

### Decision · Program_Chairs · 2023-01-20

**Decision:**

Accept: notable-top-25%

**Justification For Why Not Higher Score:**

The paper still has some points for improvement especially regarding clarity/presentation, however, a majority of the reviewers agree that the paper addresses an important, understudied problem in a reasonable way.

**Justification For Why Not Lower Score:**

The paper address an important, understudied problem in a reasonable way.

**Metareview: Summary, Strengths And Weaknesses:**

The paper proposes an approach to certify whether a given machine learning model achieves super-human performance when the dataset labels are (possibly erroneous) human annotations and not (unobserved) ground-truth labels. A majority of reviewers are in support of accepting the paper. They do provide several points for improvement that the authors should take into account when preparing the final version of the paper.

**Note From Pc:**

if the above contains the word "oral" or "spotlight" please see: "oral" presentation means -> notable-top-5% and "spotlight" means -> notable-top-25%. As stated in our emails, we are disassociating presentation type from AC recommendations